# Withaferin A: A Pleiotropic Anticancer Agent from the Indian Medicinal Plant *Withania somnifera* (L.) Dunal

**DOI:** 10.3390/ph16020160

**Published:** 2023-01-22

**Authors:** Suneel Kumar, Stephen O. Mathew, Ravindra Prasad Aharwal, Hardeep Singh Tulli, Chakrabhavi Dhananjaya Mohan, Gautam Sethi, Kwang-Seok Ahn, Kassidy Webber, Sardul Singh Sandhu, Anupam Bishayee

**Affiliations:** 1Bio-Design Innovation Centre, Rani Durgavati University, Jabalpur 482 001, India; 2Department of Microbiology, Immunology, and Genetics, University of North Texas Health Science Center, Fort Worth, TX 76107, USA; 3Department of Botany, Government College Kurai, Seoni 480 880, India; 4Department of Biotechnology, Maharishi Markandeshwar Engineering College, Maharishi Markandeshwar (Deemed to be University), Mullana-Ambala 133 207, India; 5Department of Studies in Molecular Biology, University of Mysore, Mysuru 570 006, India; 6Department of Pharmacology, Yong Loo Lin School of Medicine, National University of Singapore, Singapore 117600, Singapore; 7Department of Science in Korean Medicine, College of Korean Medicine, Kyung Hee University, Seoul 02447, Republic of Korea; 8College of Osteopathic Medicine, Lake Erie College of Osteopathic Medicine, Bradenton, FL 34211, USA

**Keywords:** withaferin A, cancer, apoptosis, angiogenesis, chemoresistance, formulations

## Abstract

Cancer represents the second most deadly disease and one of the most important public health concerns worldwide. Surgery, chemotherapy, radiation therapy, and immune therapy are the major types of treatment strategies that have been implemented in cancer treatment. Unfortunately, these treatment options suffer from major limitations, such as drug-resistance and adverse effects, which may eventually result in disease recurrence. Many phytochemicals have been investigated for their antitumor efficacy in preclinical models and clinical studies to discover newer therapeutic agents with fewer adverse effects. Withaferin A, a natural bioactive molecule isolated from the Indian medicinal plant *Withania somnifera* (L.) Dunal, has been reported to impart anticancer activities against various cancer cell lines and preclinical cancer models by modulating the expression and activity of different oncogenic proteins. In this article, we have comprehensively discussed the biosynthesis of withaferin A as well as its antineoplastic activities and mode-of-action in in vitro and in vivo settings. We have also reviewed the effect of withaferin A on the expression of miRNAs, its combinational effect with other cytotoxic agents, withaferin A-based formulations, safety and toxicity profiles, and its clinical potential.

## 1. Introduction

Although there has been phenomenal progress in diagnostics and therapeutic approaches for the management of cancer, globally it remains one of the most lethal diseases of the 21st century. In recent years, natural bioactive compounds have been recognized for their efficacy in preventing cancer-causing activity and in decreasing the progression of different malignancies [1,2,3,4]. Over the years, several phytochemicals have been identified for their potential anticancer activities [5,6,7,8,9]. Withaferin A is an important natural phytochemical widely tested for its antitumor properties against various cancer models. In most studies, the effects of withaferin A to promote cell cycle arrest, angiogenesis, antimetastatic activity, and apoptotic cell death have been observed in different cancer cell lines and animal models. Withaferin A is a steroidal lactone generally obtained from *Withania somnifera* (L.) Dunal, commonly known as Indian Ginseng or Ashwagandha, is an evergreen plant that has short stems, tuberous roots, and bisexual flowers [10,11]. In addition, various parts of the plant, including the roots, leaves, stems, and flowers, have therapeutic value, and more than twenty-nine types of bioactive compounds have been extracted from the roots and leaves alone [12,13,14]. Withaferin A, in both in vitro and in vivo studies, has shown unique therapeutic properties, including antioxidant, anti-inflammatory, antibacterial, antistress, antidiabetic, antipyretic, cardioprotective, neuroprotective, and anticancer activities [15,16,17,18,19,20,21]. The production of withaferin A from natural sources has stimulated research in various areas of chemical sciences, chemical biology, biotechnology, biochemistry, microbiology, and organic chemistry. Withaferin A plays a significant role in the inhibition of several signal transduction pathways, such as Notch-1 signaling and the downregulation of pro-survival pathways, including Akt serine/threonine kinase (Akt)/nuclear factor-κB (NF-ĸB)/B-cell lymphoma 2 (Bcl-2) in colon cancer cell lines, and the suppression of the phosphorylation and translocation of suppressor of mothers against decapentaplegic 2/3 (Smad2/3) and NF-κB in cancer cells [22,23,24,25,26,27]. Several articles have previously described the anticancer effects of withaferin A [28,29,30,31]; however, the results of these articles were restricted and lacked detailed analyses of the pleiotropic antineoplastic effects of withaferin A. In this review article, we have systematically discussed the diverse anticancer activities of withaferin A to emphasize the therapeutic potential of this promising agent against different malignancies. Furthermore, the chemical structure, chemopreventive properties, antioxidant and anti-inflammatory effects, inhibitory role in angiogenesis and metastasis, and a novel formulation of withaferin A have been discussed in this review to incorporate a holistic approach for the development of withaferin A as a potent antineoplastic drug.

## 2. Structure of Withaferin A

The bioactive compound withaferin A is a steroidal lactone primarily extracted from *W. somnifera*, first isolated by Israeli chemists, Asher Lavie and David Yarden, in 1962 from the leaves of the plant. The inhibitory properties of the withaferin A molecule are due to the occurrence of an unsaturated lactone in a side chain; allylic 1° alcohol is attached at position 25 and the highly oxygenated rings A and B are attached to the other side of the compound [32,33,34,35,36]. Cytotoxicity is achieved by the presence of α,β-unsaturated carbonyl moiety of the withaferin A structure of thiol adducts [37]. Withaferin A structural studies revealed that the three positions susceptible to nucleophilic attack are the unsaturated A-ring at C3, the epoxide at position five, and C24 in the E ring (Figure 1). All these sites of withaferin A are covalently attached to the cysteine residues of protein by the Michael addition alkylation reaction, resulting in a loss of the activity of the target protein [38]. However, the C27 hydroxy group is not biologically important for withaferin A activity and can be conjugated with biotin so that it may recognize various target proteins [39,40]. Withaferin A was first isolated from the ether extract of *Withania* plant leaves and analyzed by thin-layer chromatography (TLC) and column chromatography. The initial extraction of withaferin A was performed by repeated column chromatography using the chloroform–methanol fraction of the *W. somnifera* extract and partial purification of the compound using TLC and high-performance liquid chromatography. In the end, nuclear magnetic resonance and Fourier transform infrared were used to characterize the product [41,42].

Moreover, withaferin A may also be extracted from the root of *W. frutescens*, a plant native to Europe with a high content of withaferin A in its leaves, which has shown a similar phytochemical profile to *W. somnifera* [43]. In another study, in vitro and in situ grown *W. somnifera* plants underwent morphological and phytochemical screening for the optimization of withaferin A extraction. The highest amount of withaferin A obtained under in vitro conditions was a concentration of 0.27 to 7.64 mg/g dry weight, while under in situ conditions the concentration was 8.06 to 36.31 mg/g dry weight [44]. The evaluation of structure–activity relationships in 56 withanolides using 2D and 3D coculture models revealed withanolide D (2) as a compound that has high antiproliferative activity against multiple myeloma [45].

## 3. Biosynthesis of Withanolides

Withaferin A contains five important chemical groups: an epoxide ring between carbons 5 and carbon 6, a hydroxyl group at carbon 27, a 6-membered lactone ring (E) with an α, a β-unsaturated carbonyl group, an α,β-unsaturated ketone group in ring A, and a secondary hydroxyl group at carbon 4. Several studies have shown that changing a few of the aforementioned chemical groups aids in the synthesis of bioactive semisynthetic analogs of withaferin A [46,47,48,49]. 

Various types of withanolides are synthesized through mevalonate and non-mevalonate routes often known as the isoprenoid pathway and a range of enzymes, such as cycloartenol synthase, (S)-2, 3-epoxysqualene mutase belongs to the 2–3 oxidosqualene cyclases (OCSs) gene family along with α-amyrin synthase (AAS), β-amyrin synthase (BAS), lupeol synthase (LS), and lanosterol synthase (LAS) [50,51]. Withanolides are formed biosynthetically by cycloartenol synthase (CAS). The cyclization of 2,3-oxidosqualene into cycloartenol by the enzyme CAS is a crucial step in the formation of all withanolides in *W. somnifera* plants. A step-by-step process that includes desaturation, hydroxylation, epoxidations, cyclization, chain elongation, and glycosylation, is preceded by cycloartenol molecules that serve as a precursor for the synthesis of withanolides. The biosynthesis of withanolides began with the cyclization of epoxysqualene to produce the triterpenoid compound C_30_H_50_O. 

Plants contain various types of oxidosqualene cyclases that are essential for the synthesis of sterols and many types of triterpenoids from 2,3-oxidosqualene. In the first step of mevalonate pathways, the enzyme acetoacetyl (AcAc)-CoA thiolase is involved in the condensation of two acetyl CoA molecules into AcAc-CoA to form 3-hydroxy-3-methylglutaryl-coenzyme (HMG-CoA) [52]. In the second step, AcAc-CoA is condensed with one molecule of acetyl-CoA to form HMG-CoA with the help of an enzyme HMG-CoA synthase [53]. In the third step, a nicotinamide adenine dinucleotide (phosphate)-dependent (NAD(P)H) enzyme and HMG-CoA reductase biosynthesized mevalonate (MVA) from HMG-CoA (Benveniste, 2002). MVA is converted further into isopentyl pyrophosphate (IPP) via phosphorylation and decarboxylation, which are carried out by a series of enzymes, including MVA kinase, phosphomevalonate kinase, and MVA diphosphate decarboxylase. The IPP resulting from the cytosolic MVA pathway changes into dimethylallyl diphosphate (DMAPP) (Hunter, 2007). The IPP and DMAPP are further involved in the formation of farnesyl pyrophosphate (FPP), which is the main precursor for the synthesis of triterpenoids [54,55]. 

When withanolides are produced by the methylerythritol 4-phosphate (MEP) route, an enzyme 1-deoxy-D-xylulose 5-phosphate synthase converts the major substrates, pyruvate, and glyceraldehydes 3-phosphate, into 1-deoxy-D-xylulose 5-phosphate (DXP) [56,57]. In the next step, the DXP changes into the MEP and it further changes into the 1-hydroxy-2-methyl-2-€- butenyl 4-diphosphate, with the help of enzymes DXP reductoisomerase and successive action of other enzymes, such as 2-C-methyl-D-erythritol 4-phosphate cytidylyltransferase, 4-diphosphocytidyl-2-C-methyl-D-erythritol kinase, 2-C- methyl-D-erythritol 2,4-cyclodiphosphate synthase, and (E)-4-hydroxy-3-methylbut-2-enyl diphosphate synthase (HMBPP). The last step of this pathway involved the branching of HMBPP to IPP and DMAPP catalyzed by an enzyme, (E)-4-hydroxy-3-methyl but-2-enyl diphosphate reductase. The IPP further leads the development of the main precursor of FPP for triterpenoids biosynthesis [54,55]. The FPP reaction is promoted into the condensation of IPP with DMAPP to form 10-C intermediate geranyl diphosphate (GPP), and the condensation of GPP with IPP results in 15-C FPP [58]. Squalene, which is produced by the action of the enzyme squalene synthase, which catalyzed the condensation of two molecules of FPP in NADPH, is one of the most important molecules for the biosynthesis of triterpenoids. Now that the squalene has undergone epoxidation to create squalene 2,3,-epoxide, cycloartenol has been biosynthesized, and it has subsequently evolved into a range of different kinds of steroidal triterpenoidal compounds [58,59,60].

## 4. Derivatives of Withaferin

The extracts of withaferin derivatives have been widely used for the treatment of cancers, but the first noted anticancer action of a purified withanolide was observed in 2004 [61]. Withaferin A is the most important withanolide, and it has unique anticancer and antitumor pharmacological activities; therefore, it has a substantial potential for drug development [28,38,62,63]. Currently, 900 types of withanolides have been discovered and isolated from Solanaceae family plants, with some withaferin derivatives also extracted and characterized from other families, such as the Fabaceae, Lamiaceae, Leguminosae, Myrtaceae, and Taccaceae families, as well as the marine Alcyoniidae family. The differences in the chemical structure of withanolides have received noteworthy attention for their diverse biological effects in vitro and in vivo against cancer, inflammation, stress, microbes, and neurodegenerative conditions. Approximately 900 withanolides have been found in the last 50 years, with 24 different structural variations [64,65]. Several withanolides, including chantriolide A, physalins A, B, C, and O, viscosalactone B, withanolides D, E, and F, and withaferin A-diacetate, a diacetyl derivative of withaferin A, show anticancer properties against a variety of cancer cells [66,67,68]. Withaferin A diacetate, a derivative of withaferin, significantly decreased the viability of breast cancer stem cells [69]. Tubocapsenolide A (TA), a new withanolide derivative, can decrease the activity of the Hsp90-Hsp70 chaperone complex via diol oxidation, causing the instability of Hsp90 client proteins, and cell cycle arrest and apoptosis in human breast cancer MDA-MB-231 cells [70]. Another important derivative, W-2b, induced premature senescence associated with galactosidase activity, G2/M cell cycle arrest, and increased phosphorylation of the checkpoint kinase-2 (Chk2) in cancer cells, indicating that the W-2b derivative can inhibit tumor growth in a carcinoma model [71]. Withaferin A has a high-binding affinity and actively interacts with the key amino acid residues, acting as a significant inducer of oxidative stress via the γH2AX mechanism, while withaferin N has a low-binding affinity to amino acids, but induces cell cycle growth arrest in cancer cells while being safe to normal human cells, according to molecular docking approaches [72]. Sarkar et al. [73] observed that a withaferin derivative, Wi-D, induced apoptosis in pancreatic cancer cells by damaging β-catenin, which is involved in organogenesis and oncogenesis. Chantriolides D and E, two unique withanolides derived from the *Taccachantrieri*, were also examined for in vitro cell cytotoxicity against a tumor cell line [74]. The cytotoxicity activity of withanolides and physalins B, D, F, and U, isolated from *Physalis angulate*, were observed for their cytotoxic activity against 1A9, A431, A549, DU-145, HCT-8, HCT-116, KB, KB-VIN, LNCAP, PC-3, and ZR751 human cancer cell lines [75,76,77,78]. Withaferin A, withanone (Wi-N), and caffeic acid phenethyl ester (CAPE) have the potential to block constitutive autophosphorylation of serine/threonine-protein kinase and B-raf, which plays a significant role in regulating cell division and proliferation through the MAPK/ERK pathway [79,80,81,82]. Molecular docking and molecular dynamics simulations identified BCR-ABL, which, when constitutively activated, yields uncontrolled proliferation and suppression of apoptosis in chronic myeloid leukemia (CML), as a target for withaferin A and withanone (Wi-N) that warrants further investigations [83].

## 5. Antineoplastic Regulation of Cellular Processes in Different Tumors by Withaferin A

Withaferin A has been reported to inhibit mitochondrial fusion and reduce mitochondrial volume, leading to a lower regulation of mitofusin1, mitofusin2, and complete optical atrophy protein1 (OPA1) while also decreasing the mitochondrial fission regulation protein dynamin-related protein1 (DRP1) and the levels of OPA1 in breast cancer cells [84]. Previously, withaferin A-mediated reactive oxygen species (ROS) initiation was noticed in MCF-7 and MDA-MB-231 cell lines that inhibited the effect of withaferin A on cell death, ER-related proteins, paraptotic vacuole formation, X-box-binding protein-1, and mRNA splicing [85]. In another research, withaferin A suppressed cell proliferation in prostate, ovarian, breast, gastric, leukemic, and melanoma cancer cells and osteosarcomas by stimulating the inhibition of the cell cycle at several stages, including G0/G1 [86], G2, and M phase [87,88,89,90,91].

Withaferin A induced cell cycle arrest via the upregulation of phosphorylated Aurora B, H3, p21, and Wee-1, and the downregulation of A2, B1, and E2 cyclins, Cdc2 (Tyr15), phosphorylated Chk1, and Chk2 in DU-145 and PC-3 prostate cancer cells. These findings resulted from the activation of Cdc2 that induces M-phase cell cycle arrest, unusual duplication, and mitotic catastrophe, resulting in cell death [92]. In another study, withaferin A initiated cell death in the leukemia cells by increasing the expression of p38 mitogen-activated protein kinases (MAPK). These findings suggested that the activation of Cdc2 induces M-phase cell cycle arrest, aberrant duplication, and mitotic catastrophe, leading to substantial cell death. Moreover, RNAi p38 MAPK knockdown inhibited p38 MAPK phosphorylation, Bax expression, caspase-3 activation, and an increase in Annexin V, thus protecting leukemic cells from apoptosis. The expression of p38 MAPK in leukemia cells initiated withaferin A-mediated apoptosis by increasing the Bax levels in response to MAPK signaling, thereby leading to mitochondrial death cascades [93]. These findings provide strong evidence that withaferin A has the potential for use in the therapy of lymphoid and myeloid cells [94].

Withaferin A also significantly reduced the expression of human papillomavirus E6/E7 oncogenes in cervical cancer cells and restored the p53 pathway causing the apoptosis of cervical cancer cells. This study suggested that withaferin A-treated cervical cancer cells induced p53 accumulation, decreased the expression of HPV E6 and E7 oncoprotein, increased p21 level, and modulated the expression of proliferating cell nuclear antigen (PCNA) that triggered cell cycle arrest. In addition, the treatment led to the modulation of cyclinB1, PCNA, and p34, and altered the expression of p53-mediated apoptotic markers, such as Bcl-2, caspase-3, and PARP cleavage [95]. An oral dose of 3–5 mg/kg withaferin A attenuated the activation of Akt and stimulated Forkhead Box-O3a (FOXO3a)-mediated prostate apoptotic response-4 (Par-4) activation, effectively inhibiting the tumor progression in prostate cancer cells [96,97,98]. Several studies have also shown that colorectal adenocarcinoma cells treated with withaferin A significantly delay the mitotic cell cycle process by promoting the degradation of mitotic arrest deficiency 2 (Mad2), Cdc20, and the spindle assembly checkpoint (SAC) [99].

Withaferin A also caused apoptosis in Ca9–22 cancer cells by promoting G2/M cell cycle arrest, the generation of reactive oxygen species, histone H2AX phosphorylation, and mitochondrial membrane depolarization, indicating that withaferin A can cause the oxidative stress-mediated killing of oral cancer cells [100]. In another study, withaferin A inhibited the expansion of MCF-7 and MDA-MB-231 human breast cancer cells by ROS production, owing to mitochondrial respiration inhibition [101].

Moreover, a combination treatment of withaferin A and hyperthermia induced the death of HeLa cells via a decrease in the mitochondrial transmembrane potential and the downregulation of the antiapoptotic protein myeloid-cell leukemia 1 (MCL-1). It also induced a significant elevation in c-Jun N-terminal kinases (JNK) phosphorylation and decreased the inactivation of ERK. All the above findings indicate that withaferin A can enhance hyperthermia-induced apoptosis by stimulating mitochondria-caspase-dependent pathways [102]. Withaferin A also attenuated the development of glioblastoma multiforme (GBM), both in vitro and in vivo, by inducing endoplasmic reticulum stress via activating the transcription factor 4-ATF3-C/EBP homologous protein (ATF4-ATF3-CHOP) axis, which would be essential for optimizing withaferin A-based therapy to treat GBM by activating apoptosis and cell cycle inhibition at the G2/M level [103].

In another report, the effect of withaferin A on apoptosis has been observed in three different human colon cancer cells, through modulating the Notch-1 signaling pathway and the downregulation of Akt/NF-κB/Bcl-2 [104]. Withaferin A is also responsible for the induction of apoptosis in human breast cancer and colon cancer cells by the inhibition of cell invasion and migration by affecting the activation of the signal transducer and activator of transcription 3 (STAT3) transcription activator and ROS generation [105]. STAT3 is a transcription factor that modulates the expression of genes involved in the promotion of cell growth, survival, antiapoptosis, and metastasis [106,107,108,109]. Deregulated activation of STAT3 is frequently observed in solid and liquid malignancies, which contributes to the aggressiveness of tumors [110,111,112,113]. Withaferin A also constitutively inhibited interleukin-6-induced phosphorylation of STAT3, but not IFN-γ-induced STAT1 phosphorylation in renal Caki cancer cells. The administration of withaferin A in Caki cancer cells caused a marked downregulation of STAT3 activation and a decrease in the expression of various STAT3-regulated genes [114]. Furthermore, withaferin A treatment also led to a substantial blockade of TWIK-related acid-sensitive K^+^ channels (TASK3) in TASK3-expressing HEK 293 cells, which were concentration-dependent on the potassium current and independent of the voltage [115]. In MCF-7 and MDA-MB-231 human breast cancer cells, withaferin A displayed antiproliferative effects by causing mitotic arrest and an increase in the G2/M fraction, lowering the levels of cyclin-dependent Cdk1, Cdc25C, and Cdc25B proteins, resulting in the accumulation of tyrosine 15 phosphorylated CdK1 [116]. 

The chemopreventive effects of withaferin A were also observed in Akt-driven prostate tumorigenesis in the PTEN conditional knockout mouse model of prostate cancer, by administering the oral withaferin A at two different doses over 45 weeks. Moreover, oral administration of withaferin A resulted in significant inhibition of the tumor growth in the prostate cancer model, due to a complete absence of metastatic lesions and downregulation of p-Akt expression, β-catenin, N-cadherin and epithelial to the mesenchymal transition (EMT) markers [24].

In A549 and H1299 non-small cell lung cancer (NSCLC) cell lines, withaferin A repressed production and induced apoptosis of A549 cells by suppressing the initiation of the PI3K/Akt pathways and the phosphorylation and nuclear translocation of Smad2/3 and NF-κB, inhibiting the EMT induction in NSCLC cells [117]. In addition, others have shown that withaferin A can also exert cytotoxic effects on AGS cells by causing cell cycle inhibition at the G2/M phase and stimulating the expression of apoptotic proteins [90]. Moreover, withaferin A also promoted apoptosis in DU-145 and PC-3 prostate cancer cells and displayed fewer toxic effects against normal human fibroblasts (TIG-1 and KD). In PC-3 and DU-145, withaferin A also amplified the mRNA expression of c-FOS and 11 heat-shock proteins (HSPs) and decreased the activity of the antiapoptotic protein c-FLIP (L). It also caused the breakdown of the vimentin cytoskeleton and produced ROS in PC3 and DU-145, thereby suggesting that multiple mechanisms can mediate withaferin A-driven cell apoptosis [118].

Withaferin A and staurosporine acted as potent inhibitors of protein kinase C (PKC), this inhibition observed in *Leishmania donovani* caused the depolarization and production of ROS, which led to the release of cytochrome c into the cytosol, the initiation of oligonucleosomal DNA cleavage and caspase-like proteases, and the stabilization of the topoisomerase-1-mediated cleavage complex. All these findings indicated that PKC inhibition and stabilization of the topoisomerase-1 DNA complex by withaferin A might be essential to induce apoptotic progression [119,120,121]. An alternative mechanism through which withaferin A causes apoptosis in human colorectal cancer (CRC) cells, is through stimulating ROS production and decreasing the mitochondrial membrane potential associated with mitochondrial dysfunction. These effects were actively shown to stimulate cell apoptosis by ROS-mediated mitochondrial dysfunction and the JNKs pathway that significantly inhibited tumor cell growth [122]. Withaferin A also inhibited the binding of the transcription factor NF-ĸB to DNA and stimulated the activation of caspase-3 [123]. The combinatorial effect of withaferin A and sulforaphane was also observed in MDA-MB-231 and MCF-7 breast cancer cells, with a dramatic reduction of the expression of the antiapoptotic protein Bcl-2 and an increase in the pro-apoptotic Bax level, thus promoting cancer cell death [124]. It also downregulates the levels of cyclin D1, CDK4, and pRB, and upregulates the levels of E2F mRNA and tumor suppressor p21, independently of p53, leading to epigenetic modifications in the regulation of cancer cell senescence [125]. In another report, withaferin A inhibited the Notch signaling pathway and downregulated Akt/NF-ĸB/Bcl-2 and the expression of rapamycin signaling elements PS6K and p4E-BP1 in HCT-116, SW-480, and SW-620 colon cancer cells [126]. Withaferin A also mediated the downregulation of recombinant human growth-arrest-specific protein 6 (rhGas6) and Ax1 signaling pathways that can inhibit cell migration and initiate apoptosis in cancer cells [127]. The antiproliferative effects of withaferin A in human hepatocellular carcinoma (HCC) cells caused G1-phase cell cycle arrest with upregulation of p53, p21, as well as Bax, and downregulated the activation of Bcl-2, CDK2, and cyclin D1 [128]. Withaferin A also shows anticancer effects against U266B1 and IM-9 human myeloma cells by the induction of apoptosis associated with upregulation of Bax and cytochrome c, downregulation of Bcl-2, and activation of PARP, caspase-3, and caspase-9 cleavage [129].

Although withaferin A is a well-recognized biomolecule for promoting ROS formation in cancer cells, numerous investigations have revealed that it may initiate the indirect production of ROS, rather than being directly involved in it. According to a dual-modulated hypothesis, the withaferin A binding with Keap1 causes an increase in the nuclear factor erythroid 2-related factor 2 (Nrf2) protein levels, which in turn, regulates the expression of antioxidant proteins that can protect the cells from oxidative stress. Withaferin A can also target different antistress proteins and enhance ROS levels in aerobic metabolism [130]. The increase in ROS has the potential to stimulate the antioxidant pathway by causing a ROS imbalance, and this cytoprotection can ultimately determine the fate of the cancer cells. As a result, several ROS-dependent mechanisms have been described that can induce cell death through apoptosis, ferroptosis, and paraptosis, as shown in Figure 2 [131]. 

Mckenna et al. [132] found that withaferin A decreased the activation of NF-κB and attenuated the level of proteins engaged in B-cell receptor signaling and regulation of the cell cycle. Additionally, withaferin A increased the expression level of Hsp70 by inhibiting Hsp90 protein activity in murine and human B-cell lymphoma cell lines. Another study also demonstrated the inhibition of Hsp90 by withaferin A in a pancreatic cancer cell line through the degradation of Akt, cyclin-dependent kinase 4 Cdk4, and a glucocorticoid receptor of Hsp90 client protein by an ATP-independent mechanism. This significant finding elaborated the in vivo anticancer properties of withaferin A against pancreatic cancer [104]. Therefore, various antineoplastic properties of withaferin A, as presented in Table 1 and Table 2 [110,111,112,113,114,115,116,117,118,119,120,121,122,123], along with some other observations, provide strong evidence of the multifaceted anticancer properties of withaferin A, marking the therapeutic importance of this plant-based natural compound.

## 6. Anti-Inflammatory and Antioxidant Activities of Withaferin A

The uncontrolled regulation of various inflammatory markers, such as chemokines and cytokines, plays a key role in the inflammatory process [161,162,163,164,165,166]. Inflammation results in the release of many free radicals, including reactive oxygen species (ROS) and reactive nitrogen species (RNS), which cause oxidative stress and contribute to the development of various pathological conditions, such as atherosclerosis, cardiovascular disease, and cancer [167,168,169,170,171,172,173]. Cancer growth, progression, and chemotherapy treatment can all induce inflammation in cancer patients [174,175,176,177,178]. Therefore, as compared to synthetic drugs and formulations, natural bioactive metabolites that target different kinds of cancer may substantially alleviate side effects and offer new alternatives to the standard of care for cancer patients [179,180,181,182]. Currently, only a few bioactive compounds isolated from various natural sources have been tested clinically for cancer treatment, and one of these bioactive compounds, withaferin A, isolated from *W. somnifera*, has potential anticancer, anti-inflammatory, and antioxidant properties [183,184,185].

Withaferin A has shown cytotoxic properties against various cancers, such as leukemia, liver, oral, colon, pancreas, prostate, breast, ovarian, and bladder cancer [186,187]. The anti-inflammatory qualities of withaferin A are specifically attributed to its inhibition of pro-inflammatory molecules, α-2 macroglobulin, NF-κB, activator protein 1 (AP-1), and cyclooxygenase-2 (COX-2) inhibition, observed in various in vitro models [188]. Human and mouse islets treated with withaferin A demonstrated the inhibition of NF-ĸB signaling, preventing cytokine-induced death by reducing the secretion of cytokines, thereby effectively protecting the islet [189] (Figure 3).

In another study, it was observed that the activation of the Toll-like receptor 4 (TLR4) in spinal cord astrocytes triggers a signal flow that leads to the commencement of NF-ĸB, which further initiates the expression of TNF-α, COX-2, and inducible nitric oxide synthase (iNOS), and pro-inflammatory and stress-response moderators that may cause CNS disorders, such as neural cell death. However, withaferin A is highly effective in inhibiting the transcriptional activity of NF-ĸB and pro-inflammatory and stress-response mediators in astrocytes, exhibiting the potential for withaferin A to combat neurodegenerative disorders [190,191,192,193,194]. Other studies also revealed that withaferin A reduced NADPH oxidase as well as superoxide levels, which prevented the aging-induced neurodegeneration of the dopaminergic neurons in the brain of a rat model [195,196].

Withaferin A also acts as a key mediator in the prevention of inflammation during chronic kidney disease (CKD), seen in the unilateral urethral obstruction (UUO) renal injury animal model (unilateral obstruction). The levels of TGF-β and downstream signaling molecules p-Smad2, p-Smad3, total Smad4, p-Akt, and p-ERK were attenuated by withaferin A, showing strong evidence of the renoprotective potential of withaferin A due to its anti-inflammatory activity [197]. Withaferin A can also display antioxidant effects in liver fibrosis, by attenuating the BB-(PDGF-BB) platelet growth factor and promoting PDGF-BB-induced SIRT3 expression and action in the case of JS1 cells. It also prevented carbon tetrachloride (CCl_4_)-induced liver damage, fibrosis, and collagen deposition by increasing the sirtuin3 (SIRT3) expression and suppressing CCl_4_-induced oxidative stress in the fibrotic liver of C57/BL6 mice [198]. Studies have shown that withaferin A is capable of restoring the structure of the liver by increasing antioxidant action in hepatocarcinogenic rats by lowering the level of liver marker enzymes and reducing the oxidative stress of various oxidants [199]. Another study observed that withaferin-A activated LXR-α, which inhibits NF-κB transcriptional activity and suppresses the proliferation, invasion, migration, and anchorage-independent growth of hepatocellular carcinoma (HCC) cells, confirming withaferin A to be a potent anticancer compound that suppresses various angiogenesis and inflammatory markers linked to the development and progression of HCC.

Interestingly, low concentrations of withaferin A treatment for 24 h did not show cytotoxicity against Ca9–22 oral cancer cells, but did cause the release of ROS, wound healing, and the migration of cells. At the molecular level, withaferin A inhibits matrix metalloproteinase-2 (MMP-2) and MMP-9, but also provokes mRNA stimulation for a set of antioxidant genes, such as NADPH quinone dehydrogenase 1 (NQO1), glutathione-disulfide reductase (GSR), Nrf2, heme oxygenase 1 (HMOX1), and induced mild phosphorylation in the MAPK family, including extracellular signal-regulated kinases 1/2 (ERK1/2), c-Jun N-terminal kinase (JNK), and p38 expression in Ca9–22 cells. All these alterations were suppressed by the presence of the ROS scavenger *N*-acetylcysteine (NAC), suggesting that low concentrations of withaferin A can maintain potent ROS-mediated antimigration and invasion capabilities of oropharyngeal squamous cancer cells [200]. Withaferin A improved the ability of H9c2 cells to survive against simulated ischemia/reperfusion (SI/R) or hydrogen peroxide (H_2_O_2_)-induced cell death in a cardiac ischemia-reperfusion injury model. Withaferin A triggered the upregulation of superoxide dismutase SOD_2_, SOD_3_, and peroxiredoxin 1(Prdx-1). Additionally, withaferin A inhibited the H_2_O_2_-induced upregulation of SOD_2_, SOD_3_, and Prdx-1, and ameliorated cardiomyocyte caspase-3 activity via an Akt-dependent pathway [201].

Chaudhary et al. [202] reported that an overproduction of ROS accelerated by withaferin A was responsible for the inhibition of the cell cycle in CRC cells and that it decreased the potential of the mitochondrial membrane, causing mitochondrial dysfunction. Withaferin A promoted radiation-induced apoptosis in human kidney carcinoma (Caki) cells by producing reactive oxygen species and inhibiting Bcl-2 and Akt dephosphorylation [16]. Withaferin A, in combination with doxorubicin (DOX), is also responsible for the excessive generation of ROS that can cause concentration-dependent DNA damage and stimulate autophagy in an ovarian cancer cell line (Figure 4). The histochemical observation of tumor tissues treated with withaferin A and DOX showed a reduction in the cell proliferation and micro-vessel development, an increase in light chain 3β (LC3B) levels, DNA destruction, and the cleavage of caspase-3 [203,204,205].

## 7. Angiogenesis and Metastasis

The onset of epithelial-to-mesenchymal transition (EMT) promotes the growth of invasive and migrating tumor cells that increase metastasis [203,204,205,206]. Several studies have investigated the possible inhibitory effects of withaferin A in metastasis, migration, and invasion in various cancers [207]. Along with the initiation of apoptosis and cell cycle arrest in cancers, withaferin A has also been extensively observed to alter angiogenesis and metastasis, which are important hallmarks of cancer. The present evidence supported the fact that withaferin A acts as a strong angiogenesis inhibitor in vascular endothelial cells by targeting the ubiquitin-protease pathway. Withaferin A can effectively inhibit cell proliferation in human umbilical endothelial vein cells (HUVECs) through interference with the ubiquitin-mediated proteasome pathway, as evidenced by the elevated levels of poly-ubiquitinated proteins (Figure 5). Withaferin A was also found to exert potent antiangiogenic activity in vivo at significantly low doses [208].

In an Ehrlich ascites tumor (EAT) model, mice treated with withaferin A displayed marked inhibition of angiogenesis and micro-vessel density compared to untreated animals. The mechanism of action postulated that withaferin A decreased the binding of the transcription factor specificity protein 1 (Sp1) to VEGF to exert its antiangiogenic activity [209]. In another study, withaferin A treatment in a liver cancer nude mouse model decreased macrophage infiltration and the inhibition of protein tyrosine kinase-2 (Pyk2), Rho-associated kinases 1(ROCK1), and vascular endothelial growth factor (VEGF) expression, along with cancer tissue necrosis and actin suppression. Withaferin A appears to be a promising candidate in liver cancer therapy, due to its inhibition of tumor invasion and angiogenesis by downregulating the cell signaling pathway [210]. Moreover, in a CRC in vivo model, oral administration of withaferin A extensively suppressed Akt and its pro-survival signaling molecules along with the inhibition of EMT markers, such as Snail, Slug, β-catenin, and vimentin. This suggests that withaferin A may be able to counteract Akt-induced cell proliferation and the tumor development of colon cancer. Furthermore, withaferin A inhibited micro-vessel formation that was linked with low expression of the angiogenic marker reticulocyte (RETIC) in CRC cells [96]. In another study, high expression of vimentin in glioblastoma patients was associated with poor progression-free survival, and withaferin A inhibited glioblastoma cell migration and invasion activity. Studies have shown that vimentin enhances triple-negative breast cancer (TNBC) aggressiveness and resistance to chemotherapeutic agents, and withaferin A, an inhibitor of vimentin [211], could be a major player in combating drug resistance and the recurrence of TNBC [212].

Withaferin A treatment on human metastatic cancer CasKi cells caused the inhibition of the transforming growth factor (TGF-β)-induced expression of MMP-9. Specifically, the downregulation of MMP-9 was a result of the TGF-β-stimulated phosphorylation of Akt, and this was partially restored by introducing constitutively active (CA)-Akt, suggesting that withaferin A was able to reduce the invasive and migratory abilities of CasKi cells through a reduction in MMP-9 expression, via decreased Akt signaling [213]. In a breast cancer metastasis in vivo model, withaferin A displayed dose-dependent inhibition of metastatic lung nodules and induced vimentin ser56 phosphorylation with a very low toxic effect on the lung tissue [214]. Xu et al. [26] showed that withaferin A blocked TGF-β-dependent Smad2 phosphorylation and expression of other TGF-β-related proteins in human endometrial cancer cells, suggesting that withaferin A inhibits the proliferation of human endometrial carcinoma via TGF-β signal regulation.

In another study, the treatment of orthotopic ovarian tumors in combination with withaferin A and cisplatin helped to decrease tumor formation and inhibit metastasis to other organs. Withaferin A was highly effective in eliminating cancer stem cells (CSC) that expressed cell surface markers, such as CD24, CD34, CD44, CD117, and Oct4 while downregulating *Notch1*, *Hes1*, and *Hey1* genes; however, mice treated with cisplatin alone experienced the opposite effects on the cells [215]. In another study [202], withaferin A was found to bind strongly to vimentin and heterogeneous nuclear ribonucleoprotein hnRNP-K, and downregulate the expression of MMPs and VEGF, as well as reduce vimentin, N-cadherin cytoskeleton proteins, and protease u-PA involved in the cancer cell metastasis.

Withaferin A also decreased invasion and gene expression of extracellular matrix-degrading proteases, the pro-inflammatory mediators of the metastasis-promoting tumor microenvironment, such as tumor necrosis factor ligand superfamily member 12 (TNFSF12), interleukin-6 (IL-6), angiopoietin-like protein 2 (ANGPTL2), colony-stimulating factor-1 receptor (CSF1R), and also decreased the expression of cell adhesion proteins, integrins, and laminins, and further increased the expression of the breast cancer metastasis suppressor gene (*BRMS1*) in MDA-MB-231 breast cancer cells [62]. The limited expression of the VEGF signaling protein is responsible for ensuring a proper supply of oxygen to tissues when the blood supply is inadequate, but overexpression of this protein may lead to cancer [216,217,218]. A molecular docking study found that the binding of withaferin A to VEGF can attenuate the processes of both angiogenesis and metastasis [219]. In these studies, the action of withaferin A and other withanones against the inhibition of migration, invasion, and in vivo lung metastasis of HT1080 fibrosome cells was observed. Withaferin A may break down the link between the heterogeneous nuclear ribonucleoprotein-k (hnRNP-K), an RNA-binding protein, and the single-stranded DNA (ssDNA), via direct interaction with the hnRNP residue domain through hydrogen bonding and hydrophobic interaction. As a result, withaferin A can effectively inhibit the binding of hnRNP-k with ssDNA, thereby reducing the expression of downstream effectors of hnRNP-k, such as VEGF, PERK, and MMP2 [220].

## 8. Regulation of microRNAs (miRNAs)

In addition to its role in cellular processes, withaferin A can also influence the expression of microRNAs. miRNAs are a class of small, non-coding RNAs that control gene regulation in many cellular processes. miRNAs regulate genes that intercede processes in tumorigenesis, viz inflammation, cell cycle regulation, stress response, differentiation, apoptosis, and invasion [221,222,223,224,225]. In an interesting study, withaferin A was reported to upregulate the expression of miR25, which in turn, upregulated the COX-2 expression, inducing an inflammatory response in rabbit articular chondrocytes [226]. In lung cancer cells, the miRNAs responsible for the inhibitory effects of withaferin A were investigated, and the treatment with withaferin A caused an upregulation of the pro-apoptotic molecules, p53 and Bax, but reduced the activity of Bcl-2. Furthermore, withaferin A inhibited the functionality of lung cancer cells by regulating the two onco-miRNAs, such as miR-10b and miR-27a, which control the expression of E-cadherin and Bax in a p35-dependent manner [227]. The limited data for the miRNA mechanisms associated with withaferin A mean that it would be premature to comment on the full effects of withaferin A in modulating the levels of different miRNAs. Hence, further research is warranted to fully elucidate miRNA regulation and the activation of miRNA by withaferin A and its effects on cancer cells.

## 9. Synergistic Effects of Withaferin A

The current strategies for treating cancer, such as chemotherapy, suffer from low survival rates, severe side effects, and the development of resistance to drugs in cancer patients [228,229,230,231,232,233,234]. Therefore, alternative approaches that exert anticancer effects, including the use of natural phytochemicals, have been exploited for pharmaceutical purposes [234,235,236]. A large number of data from in vitro and in vivo studies in various cancers have shown that withaferin A has the potential to prevent cancer from developing, due to its effective anticancer properties [237,238,239]. Activation and restoration of p53 function by withaferin A has led to cell cycle arrest and death in in vitro and in vivo cancer models, as well as causing the initiation of p53 phosphorylation at serine315 residue, thereby increasing the p53-mediated transcription of p21 cell cycle inhibition in MCF-7 cells [240]. 

Several studies have also explored the effects of withaferin A, alone or in combination with other cancer chemotherapeutic agents, such as paclitaxel, on the migration, growth, spread, and metastases of human NSCLC cells, H1299 and A549. The combination of the two bioactive compounds inhibited colony formation, invasion, and migration, and improved apoptosis in the cell [117]. Withaferin A was found to enhance oxaliplatin-induced growth inhibition and apoptosis in pancreatic cells, due to the dysfunction of the mitochondria and the inactivation of the PI3K/Akt pathway [241]. Similarly, withaferin A inhibited the lung cancer spheroid-forming capacity and reduced the growth of cancer stem cells by decreasing the action of the mTOR/STAT3 signaling pathway. The combination of withaferin A and other anticancer drugs, such as cisplatin and pemetrexed, showed synergistic results in the inhibition of the epidermal growth factor receptor (EGFR) and wild-type lung cancer cell viability, and further increased the cytotoxic effect of cisplatin [158]. The synergistic effects of withaferin A and carnosol resulted in the suppression of c-Met phosphorylation, sphere-formation, and clonogenic potential, which was accompanied by the downregulation of pluripotency-maintaining genes (oct-4 and Nanog), demonstrating their ability to target pancreatic cancer stem cells [242].

In addition, the potential effects of withaferin A, alone and in combination with liposomal preparation of DOX (DOXIL), were explored in the ovarian cancer cell line A2780 and ovarian tumor-bearing mice. The combination of DOXIL and withaferin A showed a synergistic effect by inhibiting the expression of aldehyde dehydrogenase 1 (ALDH1) and Notch 1 genes. The combination of a low dose of DOXIL (2 mg/kg) and a sub-optimal dose of withaferin A (2 mg/kg) significantly reduced tumor growth and prevented metastasis, as compared to individual treatments. The combination also reduced the tumorigenic function of the cancer stem cells and the expression of the ALDH1 protein, indicating synergistic effects and supporting the potential use of combination drugs in the treatment of ovarian cancer [243]. Recently, it was reported that the combination of withaferin A and 5-fluorouracil (5-FU) inhibited cell proliferation and endoplasmic reticulum stress, leading to cell apoptosis in colorectal cancer cells (CRC). The combination of both drugs increased the expression of ER stress sensors Bip, PERK, CHOP, ATF-4, and eLF2, and decreased cell viability by the initiation of the ER stress-mediated apoptosis and autophagy, G2M-phase cell cycle arrest, and the β-catenin pathway [244].

## 10. Formulations Based on Withaferin A

Withaferin A demonstrates a broad range of properties, including antioxidant, anti-inflammatory, and antimicrobial. Gold nanoparticles conjugated with withaferin A induced a blockage of the SKBR3 breast cancer cell line at half the concentration compared to a pure withaferin A form [245]. In combination with withaferin A and gold nanoparticles, dexamethasone was able to inhibit EMT in tumor cells, prevent metastases of mouse melanoma tumors, and reduce mortality in tumor-bearing mice [246]. The use of novel synthetic variants of niosomes as the carriers of anticancer drugs is an emerging area in oncology. Withaferin A formulated in cholesterol-based and non-ionic surfactant niosomes significantly improved the anticancer activity against HeLa cells when compared to its pure form [247].

Withaferin A-loaded poly D,L-lactic-co-glycolic acid (PLGA) nanoparticles have shown improved solubility due to their small size and enhanced bioavailability, which could be used as an effective immune modulator [248]. In the treatment of pancreatic cancer, withaferin A encapsulated with a methoxy poly(ethylene glycol) (mPEG) conjugated poly(D,L-lactide-co-glycolide) (PLGA), exhibited strong anticancer properties as compared to the free form, and it also allowed strong bonding to heat-shock proteins, reducing the expression of Akt and CDK4 proteins and causing apoptosis in cancer cells [249]. In another research study, withaferin A-loaded PLGA nanoparticles developed by solvent evaporation showed good drug loading efficiency and release, offering another alternative for treatment [250]. Dhabian and Jasim [251] synthesized a nano–zinc solution of the *W. somnifera* aqueous extract with the plant extract by itself, and found the nano–zinc solution to have more potent anticancer properties against HeLa cells than the plant extract. Using withaferin A as a natural ferroptosis-inducing agent, Hassannia et al. [252] showed that the nano-targeting of withaferin A allowed the systemic application and suppressed tumor growth, possibly due to an enhanced accumulation in the tumor tissue.

To reduce bone resorption and inflammation, mannosylated liposome encapsulated withaferin A was used against synovial macrophages, in which osteoclastogenesis production was increased after treatment with encapsulated molecules, and the inhibition of cartilage and bone erosion occurred [253]. Recently, a new approach to liposomal drug delivery systems was developed specifically for targeting angiogenic endothelial cells and CD13-positive pancreatic cancer with homing peptides (NGR). Consequently, a withaferin A encapsulated-liposomal formulation of NGRKC-16 lipopeptide caused apoptosis in CD13-positive pancreatic cells and antigenic endothelial cells, which could be used in the treatment of aggressive pancreatic cancer cells [254].

## 11. Safety and Toxicity of Withaferin A

During the review of the bioactive compound withaferin A, it was recognized that one of its most important attributes is its safety for normal cells and tissues, which allows for the continuation of this drug in clinical trials. In an in vivo study, *W. somnifera* extract containing withaferin A was orally administered to Wistar rats at a dose of 2000 mg/kg/day and had no adverse effects on the animals [255]. Moreover, withaferin A is also responsible for the reduction of acetaminophen-induced liver toxicity in mice through the stress-responsive transcription factor Nrf2. Similarly, another study found that withaferin A increased the expression of SIRT3 and suppressed CCl_4_-induced oxidative stress in the fibrotic liver of a C57/BL6 mice model, assisting in reduced liver injury, collagen deposition, and fibrosis caused by CCl_4_ [198]. Withaferin A was also shown to decrease severe cerulein-induced pancreatitis caused by inflammation and oxidative stress. The treatment of scleroderma with withaferin A initiated antifibrotic activity, which was repressed by the proinflammatory fibrosis related to TGF-β/Smad signaling and the FOXO3a-Akt-dependent NF-κβ/IKK-mediated inflammatory mechanism [256,257,258,259,260]. In addition, withaferin A has also exerted potential cytotoxicity effects at a low range (up to 5 μM) in melanoma cells, as compared to non-malignant cells [155]. Withaferin A significantly inhibited the burden of breast cancer in two different subtypes, HER2-driven breast cancer, and luminal-type breast cancer-bearing rodent models, when administered through the intraperitoneal route [261]. Most of the findings related to the general safety of withaferin A support the pharmacological and pharmaceutical studies of this natural bioactive compound. However, more research and pharmacokinetic studies of withaferin A and its derivatives are required for the development of chemotherapeutic drugs in the future.

## 12. Clinical Studies of Withaferin A

Based on impressive preclinical results, the potential of withaferin to treat a wide range of disorders is significant. However, presently there is only one ongoing clinical trial to examine the efficacy of withaferin A in cancer patients. A phase-I/II clinical trial is initiated in which the combination of DOXIL and withaferin A is examined in patients with recurrent ovarian cancer. This study aims to determine the feasibility and maximum tolerance dose of withaferin A with DOXIL and to examine the complete response, partial response, and stable disease (NCT05610735). Although there is a huge amount of preclinical data available on the anticancer potential of withaferin A, these studies are yet to substantiate its antitumor efficacy in the clinical setting. Therefore, withaferin A needs to be investigated rigorously in randomized controlled trials to understand its therapeutic efficacy against human malignancies. 

## 13. Conclusions and Future Perspectives

Over the recent decades, bioactive plant secondary metabolites have been used in the prevention and treatment of various chronic diseases. As a result, the extensive application of plant secondary metabolites has gained attention in the research community for cancer therapy. Systematic research on the bioactive compound withaferin A, isolated from the medicinal plant *W. somnifera*, has demonstrated its potent anti-inflammatory, antioxidant, and pro-apoptotic properties by modulating gene expression and signaling cascades. Preclinical evidence indicates that withaferin A holds great promise as a potential anticancer drug, but due to the lack of extensive pharmacokinetic studies, the inherent use of this compound has not been capitalized. Therefore, it is distinctly important to conduct extensive pharmacokinetic and bioavailability assessments of withaferin A. Such assessments may help optimize tolerability, absorption, bioavailability, and route of administration under physiological conditions. In addition, the toxicology profile of withaferin A remains unclear, so a full elucidation is warranted. In conclusion, withaferin A holds great promise as a therapeutic agent and may improve therapeutic outcomes in patients with diverse diseases.

## Figures and Tables

**Figure 1 pharmaceuticals-16-00160-f001:**
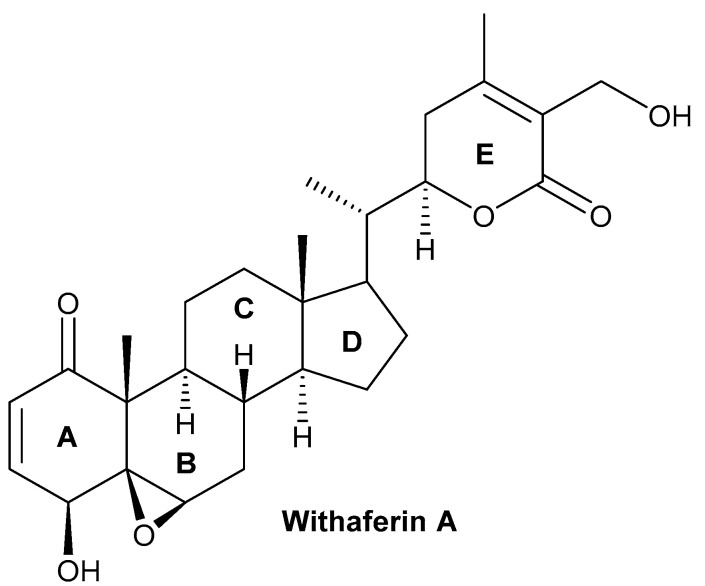
Chemical structure of withaferin A.

**Figure 2 pharmaceuticals-16-00160-f002:**
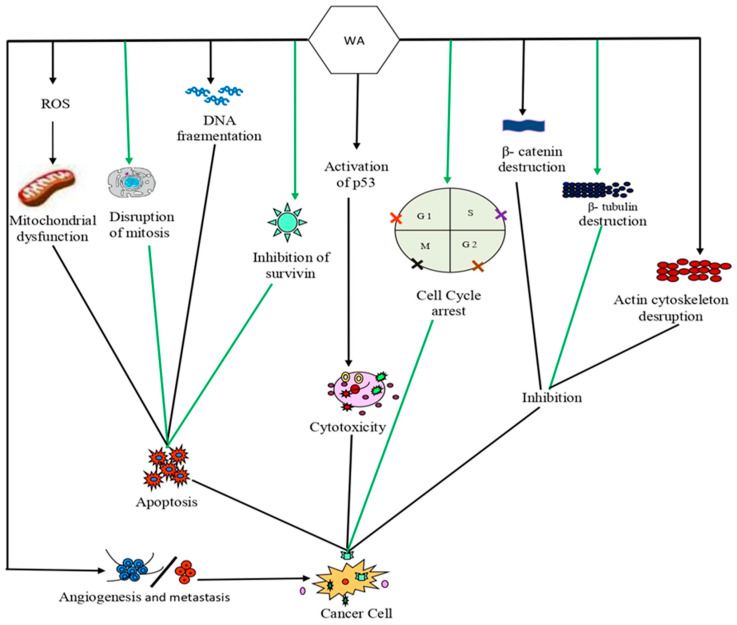
Mechanism of action of withaferin A against cancer cells: Withaferin A caused dysfunction of mitochondria and disruption of mitotic spindle assembly of cancer cells by producing excessive ROS that can lead to apoptosis. Withaferin A can also inhibit the activities of β catenin and β tubulin and can disrupt the actin cytoskeleton. Activation of p53 function through withaferin A leads to apoptosis in cancer cells. Withaferin A also increased the rate of apoptosis in cancer cells by reducing the levels of antiapoptotic genes/proteins, such as Bcl-2 and B-Bcl-xL. Abbreviations: Bcl-2, B-cell lymphoma 2; Bcl-xL, B-cell lymphoma-extra-large; ROS, reactive oxygen species.

**Figure 3 pharmaceuticals-16-00160-f003:**
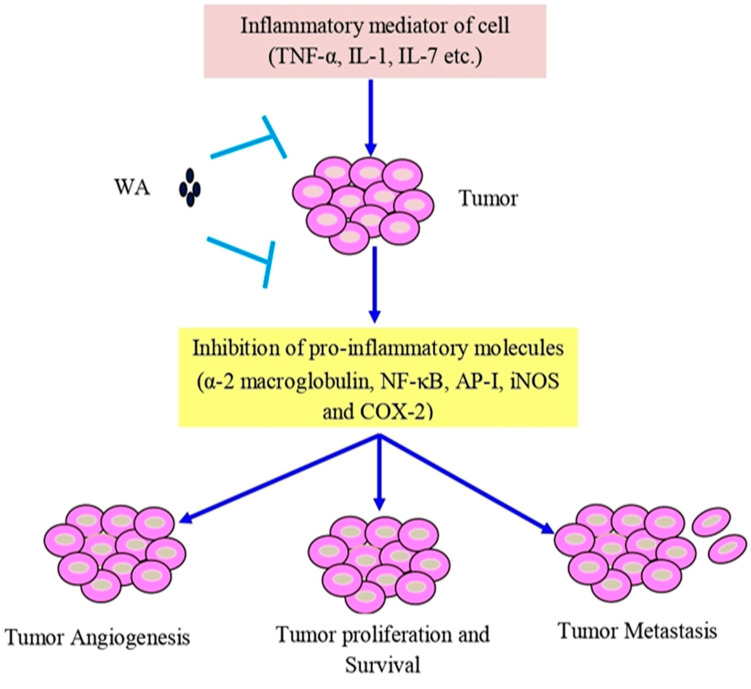
Withaferin A intercedes in anti-inflammatory function in tumor cells via TNF-α, IL-1, and IL-7. Abbreviations: IL-1, interleukin-1; IL-7, interleukin-7; and TNF-α, tumor necrosis factor-α.

**Figure 4 pharmaceuticals-16-00160-f004:**
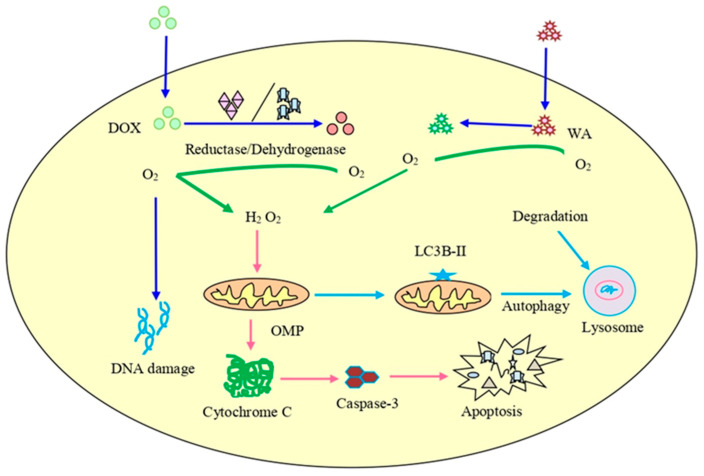
The combined treatment of doxorubicin (DOX) and withaferin A on cancer cells. It can lead to enhanced ROS production, destruction of DNA, initiation of autophagy, and increased expression of LC3B autophagy marker as well as cleavage of caspase-3.

**Figure 5 pharmaceuticals-16-00160-f005:**
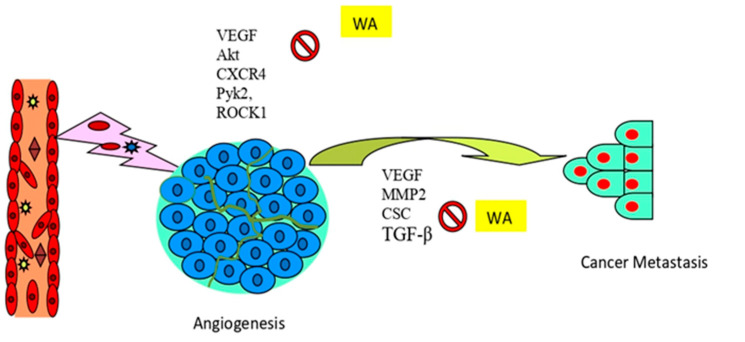
Antiangiogenic and antimetastatic properties of withaferin A. Abbreviations: CSCs, cancer stem cells; CXCR4, C-X-C chemokine receptor type 4; MMP2; matrix metalloproteinases; Pyk2, proline-rich tyrosine kinase 2; ROCK2, Rho-associated coiled-coil-containing protein kinase 2; TGF-β, transforming growth factor-β; and VEGF, vascular endothelial growth factor.

**Table 1 pharmaceuticals-16-00160-t001:** Anticancer effects of withaferin A based on selected in vitro studies.

Type of Cancer	Cancer Cell line	Concentration	Mechanism of Anticancer Activity	References
Brain cancer	U87, U251, and GL26	0.1–5 µM	Downregulated the phosphorylation of Akt, mTOR, p70S6K, and p85 S6K with increased activation of AMPKα and tuberin.	[133]
Breast cancer	MCF-7, SUM159, and SK-BR-3	1–4 µM	Directly bind to Cys (303) of β-tubulin. Decreased alpha and beta tubulin protein.	[134]
Breast cancer	MDA-MB-231 and MCF-7	0.5–4 µmol/L	Caused FOXO3a- and Bim-dependent apoptosis.	[135]
Breast cancer	MDA-MB-231 and MCF-7	1–4 µM	Enhanced ROS production, inhibited oxidative phosphorylation as well as complex III activity and activated Bax and Bak.	[136]
Breast cancer	MDA-MB-231, MCF-7, and T47D	1.25–2.5 µM	Decreased ER-α and its activity was mediated through p53.	[101]
Breast cancer	MDA-MB-231 and MCF-7	1–3 µM	Decreased Cdk1 and Cdc25B/C.	[88]
Breast cancer	MDA-MB-231 and MCF-7	2–4 µM	Inhibited constitutive/IL6-induced activation, dimerization, and nuclear translocation of STAT3.	[137]
Breast cancer	MCF-7 and MDA-MB-231	0.8–4.0 μM	Inhibited proteasome system and initiation of impaired autophagy.	[138]
Breast cancer	MCF7, MDA-MB-231, T47D, and MDA-MB-468	5 μM	Increased phosphorylation of p90-ribosomal S6 kinase and extracellular signal-regulated kinase 1/2.	[139]
Breast cancer	MCF7, MDA-MB-231, MDA-MB-468, T47D	5- 12.05 μM	Reduced proteolytic lysosomal activity with blockade of autophagic flux that inhibited growth, LDHA activity, and apoptotic induction.	[140]
Breast cancer, lung cancer, colon cancer, and brain cancer	MCF-7, NCI-H460, HCT-116, and SF-268	0.01–8.04 µg/mL	Reduced cell viability.	[141]
Breast cancer melanoma, and osteosarcoma	MCF-7, G361, and U2OS	0.25–2 µg/mL	Caused stronger telomere dysfunction and upregulated DNA damage response in telomerase-minus cancer cells.	[142]
Colon cancer	HCT116	0.0625–1 μM	Inhibited the transcriptional activity of STAT3 and suppressed migration.	[143]
Colon cancer	HCT-116, SW-480, and SW-620	4 & 5 μM	Downregulated pS6K and p4E-BP1. Inhibited Notch-mediated signaling events.	[126]
Colorectal cancer	HCT116 and SW480	0.1–10 µg/mL	Caused mitotic delay by blocking spindle assembly checkpoint function and is associated with proteasomal degradation of Mad2 and Cdc20.	[99]
Head and neck cancer	MDA1986, JMAR, UMSCC-2 and JHU011	0.1–10 μM	Stimulated apoptosis and cell death in carcinoma cells as well as cell cycle alteration from G_0_/G_1_ to G_2_/M.	[144]
Liver cancer	HepG2 and SNU449	1–100 μM	Overridden sorafenib resistance, enhanced ferroptosis, elevated Keap1, and reduced Nrf2 expression to suppress EMT.	[145]
Lung cancer	H358 and H460	10 μM	ROS-mediated cytotoxicity and apoptosis.	[146]
Lung cancer	H358 and H460	1–5 μM	Initiated apoptotic and cytostatic effect accompanied by induction of oxidative stress, increased lipid peroxidation, and GSSG/GSH ratio.	[147]
Lung cancer	A549	0.1–1 μM	Inhibited TNFα-induced expression of CAMs by inactivating Akt and NF-κB.	[148]
Lung cancer	A549	2.5–20 μM	Inhibited PI3K/Akt pathway to induce apoptosis.	[149]
Leukemia	U937	2.5–2 μM	Led to the loss of MMP, release of cytochrome c, and activated MAPK pathway.	[150]
Leukemia and myelodysplasia lymphoma	MDS92, MDS-L, HL-60, THP-1, Jurkat, and Ramos	250–1000 nM	Increased the levels of HMOX1 and LC3A/B.	[87]
Leukemia, osteosarcoma and myeloma	MOLT-4, Jurkat, REH, K562, HeLa, Saos-2, and SP2/0	1–3 μM	Activated the p38-MAPK signaling cascade and increased the phosphorylation of ATF-2 and HSP27.	[94]
Lymphoma	LY-10, LY-3, SudHL-6, Ramos, Raji, Mino, Jeko	0.1–10 μM	Induced anticancer activity likely by inhibiting Hsp90 function and NF-κB nuclear translocation.	[132]
Lymphoma	U937	0.1–1 μM	Induced PARP cleavage, activated caspase-3, and downregulated Bcl-2 in ionizing radiation-induced cells.	[151]
Oral cancer and osteosarcoma	HSC3, U2OS	4 μM	Disrupted mortalin-p53 interaction and caused the reactivation of p53 in p53^S46PΔ^ mutants.	[152]
Ovarian cancer	A2780	Dox (200 nM) plus WA (2 µM)	The combination of WA/Dox reduced cell proliferation and increased the levels of cleaved caspase-3, LCB3, and caused DNA damage.	[153]
Prostate cancer	PC-3, DU-145, and LNCaP	2–4 μM	Increased the expression of c-Fos, HSPA6, and Hsp70 and reduced expression of c-FLIP(L).	[118]
Renal cancer	Caki	4 μM	Inhibited constitutive and IL-6-induced phosphorylation of STAT3 and induces apoptosis.	[154]
Renal cancer	Caki	4 μM	Increased radiation-induced apoptosis via ROS generation, Bcl-2 downregulation, and Akt inhibition.	[16]
Skin cancer	MelCV, MelJD,	0.15–5 μM	Induced apoptosis reduced cell proliferation and inhibited migration of melanoma in cells.	[155]
Skin cancer	M14, Mel501, SK28, and Lu1205	1–12 μM	Induced apoptosis by a mitochondrial pathway with Bcl-2 downregulation, Bax mitochondrial translocation, and cytochrome c release into the cytoplasm.	[121]

Abbreviations: AMPK, AMP-activated protein kinase; ATF-2, activating transcription factor-2; CAM, cell-adhesion molecules; Cdk, cyclin-dependent kinase 1; Cdc, cell division cycle; c-FLIP(L), cellular FLICE-inhibitory protein (long form); EMT, epithelial-mesenchymal transition; ER-α, estrogen receptor-α; FOXO3a, Forkhead box class O 3a; GSH, glutathione; GSSG, glutathione disulfide; HMOX1, heme oxygenase-1; HSP, heat shock protein; IL-6, interleukin-6; Keap1, Kelch-like ECH-associated protein 1; LDHA, lactate dehydrogenase A; MAPK, mitogen-activated protein kinase; Nrf2, nuclear factor erythroid 2–related factor 2; PARP, poly (ADP-ribose) polymerase; STAT, signal transducer and activator of transcription; TNF, tumor necrosis factor; and WA/Dox, Withaferin A/doxorubicin.

**Table 2 pharmaceuticals-16-00160-t002:** Anticancer effects of withaferin A based on selected in vivo studies.

Types of Cancer	Tumor Model	Dose	Mechanism of Anticancer Activity	References
Breast cancer	MDA-MB-231 xenograft	4 mg/kg	Decreased the expression of anti-apoptotic proteins and increased apoptosis.	[139]
Breast cancer	MDA-MB-231 xenograft	4 mg/kg	Reduced the expression of survival in tumor tissues.	[156]
Breast cancer	SUM159 and MCF-7 xenograft	8 mg/kg	Downregulation of mRNA and protein level of FoxQ1 and inhibition of transcription activity.	[157]
Breast cancer	MDA-MB-231 xenograft	4 mg/kg	Reduced the levels of PCNA and TUNEL-positive cells in tumor tissues.	[135]
Colon cancer	HCT116 xenograft	2 mg/kg	Attenuated the growth of xenograft tumors in nude mice with a marked inhibition of the expression of PCNA.	[143]
Lung cancer	H441-L2G xenograft	2 mg/kg	Suppressed tumorigenesis.	[158]
Ovarian cancer	A2780 xenograft	2 mg/kg	Induced pro-inflammatory markers by ATR and attenuated Ang II level in tumor-bearing mice model.	[159]
Ovarian cancer	A2780 xenograft	Dox (1 mg/kg) plus withaferin A (2 mg/kg)	Decreased proliferation and formation of micro-vessels accompanied by an increase in LC3B level, cleaved caspase-3, and DNA damage.	[153]
Prostate cancer	PC-3 xenograft	5 mg/kg	Upregulated the expression of Par-4 and apoptosis.	[160]
Prostate cancer	PTEN-deficient Mouse	3 & 5 mg/kg	Downregulated the expression of pAkt, β-catenin, and N-cadherin.	[24]

Abbreviations: Ang II, Angiotensin II; ATR, ataxia telangiectasia, and Rad3-related protein; FoxQ1, Forkhead box transcription factor; LCB3, light chain 3B; PCNA, proliferating cell nuclear antigen; TUNEL, terminal deoxynucleotidyl transferase dUTP nick end labeling.

## Data Availability

Not applicable.

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
