# Peer review of "Withaferin A: A Pleiotropic Anticancer Agent from the Indian Medicinal Plant *Withania somnifera* (L.) Dunal"

_pharmaceuticals, 2023, doi:10.3390/ph16020160_

Round 1

Reviewer 1 Report

Dear Editor,

The authors of the above manuscript have reviewed the recent researches on the pleiotropic anticancer natural compound Withaferin A. The review is very informative and extensively constructed and I have just few comments for the authors before acceptance for publication.

1.     Overall avoid repetitions of the same concepts such as in Lane 68, 69, 74 ,351-354 or 529 where multiple times the anticancer/ant inflammatory properties have been already stated multiple times

2.     Lane 182: a reference is missing

3.     Lane 286-287: the phrase is cryptic and needs to be reformulated

4.     I would add a reference on Lane 292

5.     I would suggest the author to increase the quality and resolutions of the Figures, also for Figure 3 I would suggest to make a better picture of the different type of anti-tumor properties of WA

Author Response

The authors of this manuscript express their sincere thanks to the reviewer for the critical assessment of this work. The authors have acted upon the recommendations of the reviewer which have resulted in a significant enhancement in the quality of this manuscript. All modifications incorporated in the manuscript are highlighted in a red color font. A “point-by-point” response to each comment is outlined below.

General comments:

The authors of the above manuscript have reviewed the recent researches on the pleiotropic anticancer natural compound Withaferin A. The review is very informative and extensively constructed and I have just few comments for the authors before acceptance for publication.

Response:

We would like to thank the erudite reviewer for his/her appreciation of the manuscript. We have tried our best to address the comments raised by the reviewer and revised our manuscript.

Specific comments:

Comment 1:

Overall avoid repetitions of the same concepts such as in Lane 68, 69, 74 ,351-354 or 529 where multiple times the anticancer/ant inflammatory properties have been already stated multiple times.

Response:

We have rectified the repetitions that you have mentioned in all the specified places and in all subsequent appearances.

Comment 2:

Lane 182: a reference is missing.

Response:

As per your comment, we have added the missing reference (reference number 93) as suggested.

Comment 3:

Lane 286-287: the phrase is cryptic and needs to be reformulated.

 Response:

As per your kind suggestion, we have reformulated the suggested phrases (page 7, lines 337-339).

Comment 4:

I would add a reference on Lane 292.

Response:

As per your comment, we have added the missing reference (reference number 130).

Comment 5:

I would suggest the author to increase the quality and resolutions of the Figures, also for Figure 3 I would suggest to make a better picture of the different type of anti-tumor properties of WA.

Response:

Thank you so much for the suggestion. We have tried our best to increase the quality of the figures. In figure 3, we have attempted to present the anti-inflammatory action of the withaferin A (not anticancer mechanisms). Withaferin A demonstrated anticancer potential against a broad range of oncogenic signaling pathways as reported by hundreds of studies. Therefore, we have restricted figure 3 to the anti-inflammatory mechanism of action.

Additionally,

  1. The entire manuscript has been thoroughly checked and edited to minimize typographical errors as well as to ensure uniform style, organization, and quality.
  2. The reference list has been modified. Special attention is given to conform to the order of references and bibliographic style of the journal.

Finally,

On behalf of my co-authors, I once again express my sincere thanks to the erudite reviewer for the valuable suggestion.

Reviewer 2 Report

I have reviewed the paper entitled “Withaferin A: A Pleiotropic Anticancer Agent from the Indian 2 Medicinal Plant Withania somnifera (L.) Dunal” which contains a very systematic record. However the review needs minor modification.

-        The abstract should be a summary of the whole text.

-        The introductions should be enriched with more data from SCOPUS.

-        The resolution of all figures should be improved.

-        A detailed biosynthesis and clinical application should be included.

-        The grammatical issue should be fixed.

Author Response

The authors of this manuscript express their sincere thanks to the reviewer for the critical assessment of this work. The authors have acted upon the recommendations of the reviewer which have resulted in a significant enhancement in the quality of this manuscript. All modifications incorporated in the manuscript are highlighted in a red color font. A “point-by-point” response to each comment is outlined below.

General comments:

I have reviewed the paper entitled “Withaferin A: A Pleiotropic Anticancer Agent from the Indian 2 Medicinal Plant Withania somnifera (L.) Dunal” which contains a very systematic record. However, the review needs minor modification.

Response:

We would like to thank the erudite reviewer for his/her appreciation of the manuscript. We have tried our best to address the specific comments and revised our manuscript.

Specific comments:

Comment 1:

The abstract should be a summary of the whole text.

Response:

We appreciate this useful suggestion. We have revised the abstract to reflect on the contents of this manuscript. All changes are highlighted in red in the revised manuscript.

Comment 2:

The introductions should be enriched with more data from SCOPUS.

Response:

We sincerely appreciate this comment. We have incorporated recent information from the previous reviews/research articles and improved the introduction to withaferin A (particularly structure and biosynthesis of Withaferin A) [references 92-99 and 115-169]. All changes are highlighted in red font.

Comment 3:      

The resolution of all figures should be improved.

Response:

We agree with this suggestion. The quality of the figures has been improved.

Comment 4:

A detailed biosynthesis and clinical application should be included.

Response:

Thanks for the useful comment. We have added text on biosynthesis (page 3, line 112 to page 4, line 163) and clinical application (page 18, lines 692-703).

Comment 5:

The grammatical issue should be fixed.

Response:

We thank the reviewer for this important suggestion. We have thoroughly proof-read our manuscript and tried our best to resolve grammatical issues. All changes are highlighted in red.

Additionally,

  1. The entire manuscript has been thoroughly checked and edited to minimize typographical errors as well as to ensure uniform style, organization, and quality.
  2. The reference list has been modified. Special attention is given to conform to the order of references and bibliographic style of the journal.

Finally,

On behalf of my co-authors, I once again express my sincere thanks to the erudite reviewer for the valuable suggestion.

Round 2

Reviewer 2 Report

The authors have revised the paper as per the comments.